# Analysis of seismic strain release related to the tidal stress preceding the 2008 Wenchuan earthquake

Xuezhong Chen[1], Yane Li[1],Lijuan Chen[2]

[1]Institute of Geophysics, China Earthquake Administration, Beijing 100081,China

[2]Chongqing Earthquake Administration, Chongqing 401147,China

*Correspondence to*: Yane Li( luckystarabcd@163.com); Xuezhong Chen(cxz8675@163.com)

**Abstract.** Tidal stresses could load or unload the focal media and trigger small to moderate earthquakes in and around the focal region before a large or great earthquake. Based on the Preliminary Reference Earth Model, we calculated the time series of tidal Coulomb failure stress (TCFS) acting on the focal fault plane of the Wenchuan earthquake. For

earthquakes ($2.5 \le M_L \le 4.0$) that occurred in and around the focal region from January 1990 to April 2008, we calculated the time rate of TCFS, $\Delta$TCFS, at the occurrence time of each earthquake. These earthquakes were divided into two categories on the basis of the signs of $\Delta$TCFS: One is positive earthquakes (PEQs) occurring at times of $\Delta$TCFS >0 and the other negative earthquakes (NEQs) occurring at times of $\Delta$TCFS <0.

First, we obtained cumulative seismic strain release (CSSR) curves of NEQs and PEQs, and found that the two curves

nearly overlapped prior to September 2004 and then began to separate increasingly with time. We used a parameter $R_p$, the proportion of seismic strain release of PEQs, to investigate the effect of TCFS on earthquake occurrence, and found that $R_p$ was significantly higher than 0.5 about six months before the Wenchuan event at a 99% confidence level, indicating a significant correlation between earthquake occurrence and increasing TCFS.

Furthermore, we calculated the slope $k$ (time rate) of the CSSR curve vs. time for PEQS and NEQs respectively. It was

observed in the pre-event period that the seismic strain release accelerated when TCFS increased but decelerated when TCFS decreased. The difference in the time rate of seismic strain release between PEQS and NEQS was quantified using $R_k$, the ratio of $k$ for PEQs to that for NEQs. We discovered the stable $R_k$ values ( around 1.0) until it began to rise rapidly with time in early 2005, reaching its highest value of 2.7 just before the Wenchuan event. $R_k$ could reveal the promoting and inhibiting effects of the tidal stress on seismic strain release. When $k_p$ increases alone or $k_n$ decreases

alone , $R_k$ will increase. Thus, an increase in $R_k$ corresponds to a promoting effect during times of increasing TCFS and an inhibiting one during that of decreasing. Both effects were observed in the focal region prior to the Wenchuan mainshock.

The $b$-value in the Gutenberg-Richter relationship decreases as the tectonic stress in the crust increases. We also

calculated the temporal evolution of *b*-value in the study region. It was observed that after two and a half years of increasing tectonic stress, the focal region became unstable, and the tidal stress began to take effect as compared $R_k$ with the *b*-value. The effects of the tidal stress were gradually enhanced as the tectonic stress increased further. The increase in the tidal Coulomb failure stress may have aided the occurrence of earthquakes, whereas the decrease had the opposite effect.This observation may shed light on the seismogenic processes that led to the Wenchuan earthquake and its precursors.

**1 Introduction**

An $M_s$ 8.0 earthquake occurred in Wenchuan county, Sichuan province,China on May 12, 2008, with an epicenter at (31.0°N,103.4°E) and a depth of 19km, rupturing along the Longmenshan fault (indicated by F in Fig. 1a) . It killed thousands of people, caused damage to buildings, triggered widespread landslides, and was followed by floods and epidemic outbreaks(Yan et al. 2009; Cao et al., 2010;Zhu et al., 2012), along with serious affection of the ecological environment(Huang et al., 2018).

Scientists have reported their research on the Wenchuan earthquake, involving the co-seismic changes in water level and water temperature associated with the Wenchuan earthquake (He et al., 2016, 2017; He and Singh, 2019) , the changes in the *b*-value (Zhao and Wu, 2008; Shi et al., 2018; Chen and Zhu, 2020), the tide-triggered earthquakes (Li and Chen, 2018) and correlation between the earthquake occurrence and Earth's rotation in the pre-mainshock (Chen and Li, 2019). Meanwhile, this paper focused on the seismic strain release associated with the tidal stress prior to the 2008 Wenchuan earthquake.

The amplitude of stresses caused by the solid Earth tides in the crust is ～ 1 kPa, much lower than the average earthquake stress drop (～ $10^3$–$10^4$ kPa), and they cannot provide the energy released in earthquakes (Scholz, 2002). However, if the tectonic stress in the focal region reaches a critical value, the tidal stress could trigger an earthquake (Rydelek et al., 1992). Numerous studies have examined correlations between Earth tides and earthquakes. Positive correlation for aftershocks, volcanic earthquakes, and small to large earthquakes were obtained (Hofmann, 1961; Ryall, 1968; Shlien, 1972; Kayano, 1973; Filson et al., 1973; Mauk and Kienle, 1973; Tamrazyan, 1974; Klein, 1976; Gao, 1981; Kilston and Knopoff, 1983; Rydelek et al., 1988; Wilcock, 2001; Stroup et al., 2007; Zhang et al., 2007; Li and Jiang, 2011; Vergos et al., 2015), but there were some exceptions (Schuster, 1897; Knopoff, 1964; Shlien ,1972; Heaton, 1982; Rydelek et al., 1992; Tanaka et al., 2006). It seems that tidal triggering of earthquakes with dip-slip or oblique-slip focal mechanisms may be more significant (Heaton, 1975; Tsuruoka et al., 1995; Tanaka et al., 2002a;

Cochran et al., 2004; Li and Zhang, 2011; Bucholc and Steacy, 2016). The tidal stress triggered shallow strike-slip earthquakes that occurred in or near mainland China, but oblique-slip or dip-slip earthquakes in the same area were not triggered by tidal stresses, nor were strike-slip earthquakes occurring in California, USA (Ding et al., 1983; Vidale et al., 1998). No statistically significant evidence for a focal mechanism-dependence on earthquake tidal triggering was found in the NEIC catalog (Métivier et al., 2009). The effect of tidal Coulomb stress triggering is more significant for normal slip earthquakes at low and middle latitudes and reverse-slip earthquakes at middle and high latitudes, and the tidal stress triggering decreases with increasing latitude for strike-slip earthquakes (Xu et al., 2011). A high correlation between Earth tides and earthquake occurrence was detected in and around the epicenters in the several years prior to some moderate to large earthquakes (Chen and Ding, 1996; Chen et al., 1998; Tanaka et al., 2002b; Tanaka, 2010, 2012; Li et al., 2018).

Researchers have shown interest in the seismic strain (or moment) release acceleration near the epicentral area before strong earthquakes (Sykes and Jaumé, 1990; Bufe and Varnes, 1993; Brehm and Braile, 1998,1999; Bowman et al., 1998; Yang and Ma, 1999; Jiang et al., 2004, 2009a, 2009b,2009c; Zhang et al., 2014; Li et al., 2015; Qian et al., 2015). Although accelerating seismic strain release has been reported before some strong earthquakes; however, significant accelerating seismic strain release has not been found in some cases, even the seismic strain release decelerates. Typically, researchers investigated the accelerating seismic strain release before strong earthquakes using the method proposed by Bufe and Varnes (1993), which is based on the cumulative seismic strain release curve of small to moderate earthquakes occurring near the epicenter over a specific time period (often several years to tens of years) before the strong earthquakes and presented their findings to demonstrate whether there is a significant accelerating seismic strain release. They analyzed the shape of seismic strain release curve as a function of time by considering the studied period as a whole. The curve of seismic strain release over a longer time can be viewed as a chain of straight lines with various slopes. When the seismic strain release accelerates, the slope of the straight lines will get greater and greater, and vice versa.

We will examine whether there was any difference in the seismic strain release when the tidal stress increased and when it decreased for earthquakes that occurred before the 2008 $M_s$ 8.0 Wenchuan earthquake based on the above idea and considering the effects of the tidal stress.

## 2 Study region and data used

Earthquakes used in this study were obtained from the China Earthquake Networks Center, China Earthquake

Administration. The Wenchuan earthquake's aftershocks ($M_L \geq 3.0$) that occurred from May 12 to August 31, 2008, are

plotted in Fig. 1a. The aftershocks extended ~350 km to the northeast. A very large part of fault slip during the

occurrence of the Wenchuan mainshock took place within a region between the Maoxian county and the Dachuan town

in the southwestern aftershock zone (Zhang et al., 2008), meanwhile larger values of seismic strain release for

aftershocks from May 12 to 31, 2008 were located within the same region. This region, enclosed by a quadrangle with

a length of ~140 km in Fig. 1b, was defined as the study region in this article due to its significant correlation with the

occurrence of the Wenchuan mainshock.

The magnitudes versus time for earthquakes ($M_L \geq 2.0$) that occurred in the study region between January 1990 and

April 2008 are plotted in Fig. 2a. It can be seen that fewer earthquakes with $M_L \geq 2.0$ occurred before 2000 resulting

from the sparse seismic stations laid in and around the study region. The observed Gutenberg-Richter relationship is

usually used for determination of the threshold of completeness of earthquake catalogue via inspection. The G-R

relationships were plotted in Fig. 2b for earthquakes before and after 2000 respectively. The plot suggests the threshold

of completeness to be $Mc=2.5$ before 2000 and $Mc=1.5$ after that. It can be also found from the G-R relationship that

earthquakes with a magnitude $M_L>4.0$ does not obey the linear relationship. After we excludes those $M_L>4.0$

earthquakes, finally 217 earthquakes with a magnitude span of $2.5 \leq M_L \leq 4.0$ are used in this study.

**3 Analytical method**

Based on the Preliminary Reference Earth Model (Dziewonski and Anderson, 1981), the tide-generating stress

components in the Earth's interior are calculated. The potential due to the attraction of the moon and the Sun at the

point A($r,\theta,\lambda$) can be written as follow(Luo et al., 1986).

$$\left. \begin{array}{l} V_m(A) = \frac{3}{4} D \frac{C_m^3}{r_m} \frac{1}{R^2} \sum_{n=2}^{\infty} \left( \frac{r}{r_m} \right)^n P_n ( cos Z_m ) \\ \\ V_s (A) = \frac{3}{4} D_s \frac{C_s^3}{r_s} \frac{1}{R^2} \sum_{n=2}^{\infty} \left( \frac{r}{r_s} \right)^n P_n ( cos Z_s ) \end{array} \right\} \qquad (1)$$

Where $D$ is 26277cm$^2 \cdot$s$^{-2}$, the Doodson constant, $D_s=0.45924D$, $r_m$ is distance between the centre of the Earth and the

moon, $r_s$ is distance between the centre of the Earth and the Sun, $r$ is radius from the Earth's centre, $Z_m$ is the geocentric

zenith distance of the moon at the point A, $Z_s$ is the geocentric zenith distance of the Sun at the point A, $R$ is the Earth's

mean radius (taken to be 6371024m), $C_m$ is the average distance between the Earth and the moon, equal to 3.844×10$^8$m,

$C_s$ is the average distance between the Earth and the Sun ,equal to 1.496×10$^{11}$m, $\lambda$ is easterly longitude, $\theta$ is colatitude.

The ratial, colatitudinal and longitudinal displacements caused by the potential are given by

$$\left.\begin{aligned} u_r(A) &= \sum_{n=2}^{\infty} \frac{H_n(r)}{g(r)} V_n(A) \\ u_\theta(A) &= \sum_{n=2}^{\infty} \frac{L_n(r)}{g(r)} \frac{\partial V_n(A)}{\partial \theta} \\ u_\lambda(A) &= \sum_{n=2}^{\infty} \frac{L_n(r)}{g(r)} \frac{\partial V_n(A)}{\partial \lambda} \end{aligned}\right\} \tag{2}$$

Where $V_n = V_m + V_s$, $g(r)$ is the acceleration due to gravity. $H_n(r)$ and $L_n(r)$ are Love's numbers.

The strain components are obtained by

$$\left.\begin{aligned} \varepsilon_r &= \frac{\partial u_r}{\partial r} \\ \varepsilon_\theta &= \frac{u_r}{r} + \frac{\partial u_\theta}{r \partial \theta} \\ \varepsilon_\lambda &= \frac{u_r + u_\theta \cot \theta}{r} + \frac{\partial u_\lambda}{r \sin \theta \, \partial \lambda} \\ \varepsilon_{r\theta} &= \frac{\partial u_r}{r \partial \theta} + \frac{\partial u_\theta}{\partial r} - \frac{u_\theta}{r} \\ \varepsilon_{r\lambda} &= \frac{1}{r \sin \theta} \frac{\partial u_\lambda}{\partial \lambda} + \frac{\partial u_\lambda}{\partial r} - \frac{u_\lambda}{r} \\ \varepsilon_{\lambda\theta} &= \frac{1}{r}\left(\frac{\partial u_\lambda}{\partial \theta} - u_\lambda \cot \theta\right) + \frac{1}{r \sin \theta} \frac{\partial u_\theta}{\partial \lambda} \end{aligned}\right\} \tag{3}$$

The stress components are obtained by

$$\tau_{ij} = \lambda' \Theta \delta_{ij} + 2\mu \varepsilon_{ij} \tag{4}$$

Where $\lambda'$ and $\mu$ are Lame's coefficients, $\Theta$ is bulk strain and $\delta_{ij}$ is Kronecker operator.

According to the focal mechanism solution of the Wenchuan earthquake, the tidal stress components are projected onto its focal fault plane. The tidal normal stress $\sigma_n$ and shear stress $\tau$ can be obtained, and then the tidal Coulomb failure

stress (TCFS) acting on the focal fault plane can be obtained by applying equation (5):

$$TCFS = \tau + \mu \sigma_n \tag{5}$$

Where $\mu$ is the coefficient of friction, taken to be 0.6 (Chen, 1988). According to the global CMT catalog, the focal fault plane of the Wenchuan earthquake is a thrust-type one with the geometry of strike = 231° and dip = 35°. The rake is 138°. In calculation, the focal depth was taken to be 19 km. Fig. 3 shows the temporal variations of TCFS caused by

tides on the focal fault plane of the Wenchuan earthquake at a depth of 19 km.

We calculated the time series of TCFS at the epicenter of each earthquake. Based on the time series, we also calculated the TCFS rate (ΔTCFS) at the occurrence time of each earthquake. When TCFS increases, ΔTCFS >0 and vice versa. Earthquakes were divided into two categories: positive earthquakes (PEQs) occurring at times of ΔTCFS >0 and negative earthquakes (NEQs) occurring at times of ΔTCFS <0. Thus, the characteristics of the seismic strain released

during positive and negative TCFS can be analyzed using the above information.

In seismology, the seismic strain release $\varepsilon$ is represented by the Benioff strain obtained by taking the square root of seismic energy $E_s$ calculated from equation (6) (Gutenberg and Richter, 1956). For earthquakes in mainland China, $M_s$

in Equation (6) can be obtained from $M_L$ by Equation (7) (Fu and Liu, 1991). We arranged the earthquakes in chronological order and then obtained the cumulative seismic strain release (CSSR) versus time by accumulating their Benioff strain.

$$Log\ E_s = 1.5M_s + 4.8 \tag{6}$$

$$M_s = 1.13M_L - 1.08 \tag{7}$$

## 4 Results

Fig. 4a shows the CSSR curves of NEQs and PEQs. The CSSR curve for PEQs is represented by the grey circle, while the cyan square represents the CSSR curve for NEQs. The two curves almost overlapped before September 2004. However, they began to diverge increasingly with time afterward, indicating that the seismic strain release of PEQS was higher than that of NEQs.

The proportion of seismic strain release for PEQs $R_p$ as a function of time was calculated using a three-monthly moving 5-year time window. $R_p$ is defined as

$$R_p = \frac{\varepsilon_p}{n} \tag{8}$$

Where $\varepsilon$ is the total seismic strain release of PEQs and NEQs, and $\varepsilon_p$ is that of PEQs. Fig. 4b depicts $R_p$ vs. time. It ranged between 0.3 and 0.6 prior to October 2007, then it surpassed 0.7. As the length of time with ΔTCFS >0 is approximately equal to that with ΔTCFS<0, if the tidal Coulomb failure stress does not affect earthquakes, the normal value of $R_p$ is 0.5, and if increasing TCFS affects seismic strain release, $R_p$ should be significantly greater than 0.5, as measured by its z-values (Ge and Wang, 2006). The z-value of $N$ earthquakes can be calculated according to equation (9).

$$z = (2R_p - 1)\sqrt{N} \tag{9}$$

where $N$ is the total number of earthquakes used to calculate $R_p$. The critical $z$-value is denoted by $z_\alpha$, for which values at different significance levels are shown in Table 1. The z values for the last two $R_p$ values in Fig. 4b are 2.63 and 4.38, indicating a significant difference between the two $R_p$ values and 0.5 at a 99% confidence level. Thus, the seismic strain release was significantly related to the increasing tidal Coulomb failure stress.

Table 1 The values of $z_\alpha$ at different significance levels.

| $\alpha$ | $z_\alpha$ | | $\alpha$ | $z_\alpha$ |
|---|---|---|---|---|
| | | | | |

| 0.001 | 3.29 | | 0.01 | 2.575 |
|-------|------|---|------|-------|
| 0.002 | 3.09 | | 0.02 | 2.336 |
| 0.005 | 2.81 | | 0.05 | 1.96 |

The slope, $k$ of the CSSR curve, can represent the time rate of seismic strain release. The seismic strain release accelerates as the slope increases and vice versa. The observed slope as a function of time was obtained by fitting the data with straight lines over a 6-year time window that moved in 6-month steps. Let $k_p$ denote the slope for PEQs and $k_n$ for NEQs; both are shown in Fig. 4c using the orange circle "●" for $k_p$ and the cyan square "■" for $k_n$, respectively. The seismic strain release accelerates for PEQs when $k_p$ increases and NEQs when $k_n$ increases. $k_p$ and $k_n$ had almost the same value simultaneously and in phase before 2005. After that, they changed out of phase, and $k_p$ increased with time, whereas $k_n$ decreased. Thus, even in the several years before the Wenchuan event, the seismic strain release accelerated when the tidal Coulomb failure stress increased. At the same time, it decelerated when the tidal Coulomb failure stress decreased.

We analyzed the difference between $k_p$ and $k_n$ using their ratio $R_k$ defined as

$$R_k = \frac{k_p}{k_n} \tag{10}$$

It can be seen from $R_k$ vs. time, as shown in Fig. 4d, that $R_k$ began to rise rapidly in early 2005, reaching its peak value just before the Wenchuan earthquake. This means that, compared to NEQs, the seismic strain release rate for PEQs increased dramatically before the Wenchuan earthquake. Just before the Wenchuan event, $k_p$ reached ~2.7 fold than $k_n$. The decrease of parameter $b$ in the *G-R* relationship *log N(M) = a - bM* is interpreted as a stress increase in the crust before an upcoming seismic event (Scholz, 1968; Wyss, 1973). We investigated the temporal changes in crustal stress by $b$-value in the study region to analyze the relationship between $R_k$ and the regional tectonic stress. The maximum likelihood method was used to calculate $b$-value [Aki, 1965].

$$b = \frac{\log e}{\overline{M} - M_{min}} \tag{11}$$

The 95% confidence standard deviation of $b$ is

$$\sigma(b) = 1.96 \frac{b}{\sqrt{N-1}} \tag{12}$$

Where $\overline{M}$ represents the average magnitude of a group of earthquakes, $M_{min}$ is the minimum magnitude in the group. We calculated the $b$-value as a function of time using the earthquakes with $M_L \geq 1.5$ in the study region from January 2000 to April 2008 because fewer earthquakes occurred before 2000. Calculations of b(t) were performed in sliding

time windows with a constant number of 400 events that advanced in steps of 30 events. The red line shows the temporal changes of the $b$-value in Fig. 5d, where the grey area indicates the 95% confidence interval. In the six years leading up to the Wenchuan event, the $b$-value dropped by 31.6%, from 1.52 in May 2002 to 1.04 just before the event.

Before 2005, it decreased by 17.8% and 13.8% in the last three years and four months. Stable $R_k$ values (around 1.0) are found during the previous period of decreasing $b$-value, but when the $b$-value dropped to 1.25 at the end of 2004, $R_k$ started to rise, and the $b$-value continued to fall while $R_k$ increased rapidly, eventually reaching 2.7.

The $b$-value can reflect the regional tectonic stress, with a decrease in its value corresponding to an increase in regional tectonic stress. Therefore, $R_k$ remained stable, around 1, during the early stage of the regional tectonic stress

enhancement, indicating that TCFS did not affect seismic strain release, but $R_k$ rapidly enhanced as the regional tectonic stress increased, reaching a maximum value of 2.7 as the Wenchuan mainshock approached (see the dashed black frame in Fig. 4d). This means that the rate at which the seismic strain was released during the time of increasing TCFS was ~2.7-fold greater than that during the time of decreasing TCFS when the focal source region of the Wenchuan event was approaching instability.

To summarize the preceding observations, there was a significant stress buildup around the epicentral area preceding the Wenchuan mainshock. The difference in seismic strain release between earthquakes that occurred when TCFS increased and those occurring when TCFS decreased became increasingly noticeable during the latter phase of the stress buildup and peaked just before the Wenchuan mainshock.

**5 Conclusions and discussions**

In the present article, we examined the difference in seismic strain release between earthquakes that occurred during the increase in tidal Coulomb failure stress and those that happened during the decrease preceding the Wenchuan earthquake. The obtained results are as follows:

(1) The proportion of seismic strain released during the increase period of the tidal Coulomb failure stress was significantly greater than 0.5 at the 99% confidence level around the epicentral area about six months before the

205 Wenchuan event, indicating a significant correlation between the earthquake occurrence and increasing tidal Coulomb failure stress.

(2) For several years prior to the Wenchuan event, the seismic strain release accelerated during the increase period of the tidal Coulomb failure stress and decelerated during the decrease one.

(3) When the Wenchuan earthquake was approaching, the ratio ($R_k$) of the time rate of seismic strain release during the

increased time interval of tidal Coulomb failure stress to that during the decreased time interval increased rapidly, reaching ~2.7.

The $b$-value, which is related to the tectonic stress in the crust, had been declining since May 2002, until the Wenchuan event. By comparing ratio $R_k$ with the $b$-value, it can be found that the tidal Coulomb failure stress did not affect the seismic strain release in the early period of tectonic stress build up. However, as tectonic stress increased further, the difference in seismic strain release between NEQs and PEQs became apparent. The difference grew gradually over time, and the effect of tidal Coulomb failure stress on seismic strain release became increasingly significant.

The Earth tides produce cyclic stress variations in the Earth. These stress variations, which are of the order of $10^3$~$10^4$ Pa, are small in comparison to tectonic stresses. When the tectonic stress in a focal region is low, tidal stress does not affect earthquakes. However, when it is close to a critical condition for releasing a large rupture, the tidal stress may affect the earthquake occurrence. The increase in tidal stress promotes the earthquake occurrence, causing strain release acceleration for PEQs (corresponding to the rise in $k_p$ in Fig.4c). In contrast, the decrease in tidal stress inhibits the earthquake occurrence, causing strain release deceleration for NEQs (corresponding to the decrease of $k_n$ in Fig.4c).

It can be concluded that the increase in tidal Coulomb failure stress within three years or more before the Wenchuan earthquake might have aided the occurrence of earthquakes, whereas its decrease had the opposite effect. This observation could shed light on the processes that led to the Wenchuan earthquake and its precursors.

**Acknowledgments**

The authors express sincerely thanks to the journal editors for their help and beneficial comments to the manuscript. This study was supported by China National Key Research and Development Program (2018YFC1503400)

**Data Availability Statement**

The Earthquakes catalog support the findings of this study are available in the China Earthquake Networks Center, China Earthquake Administration at [http://10.5.160.18/console/index.action]

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

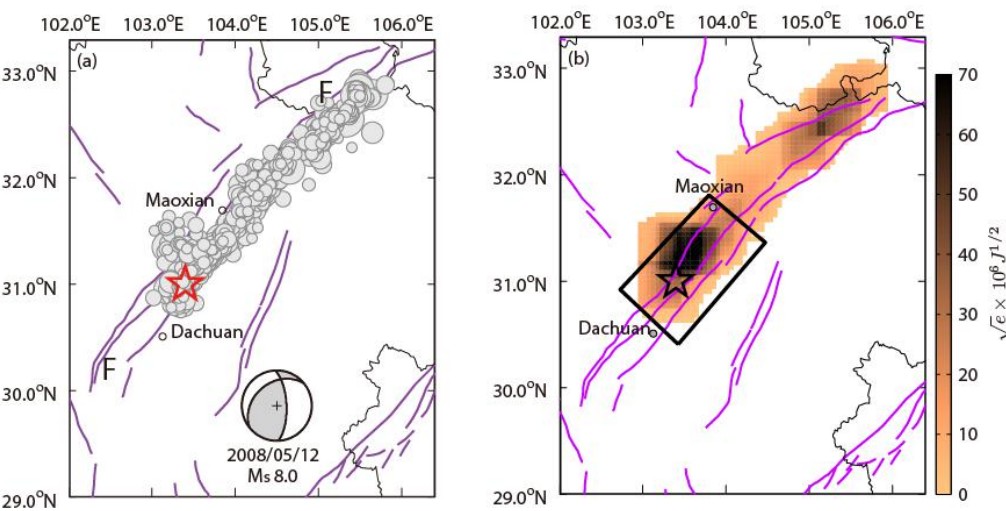

Figure 1: (a) Map showing the locations of aftershocks ($M_L \geq 3.0$) following the Wenchuan event from May 12 to August 31, 2008. The focal mechanism solution comes from the Global Centroid Moment Tensor catalog. "F" represents the Longmenshan fault. (b) The spatial distribution of seismic strain for the aftershocks that occurred from May 12 to 31, 2008. The star shows the epicenter of the Wenchuan event. The quadrangle shows the study region.

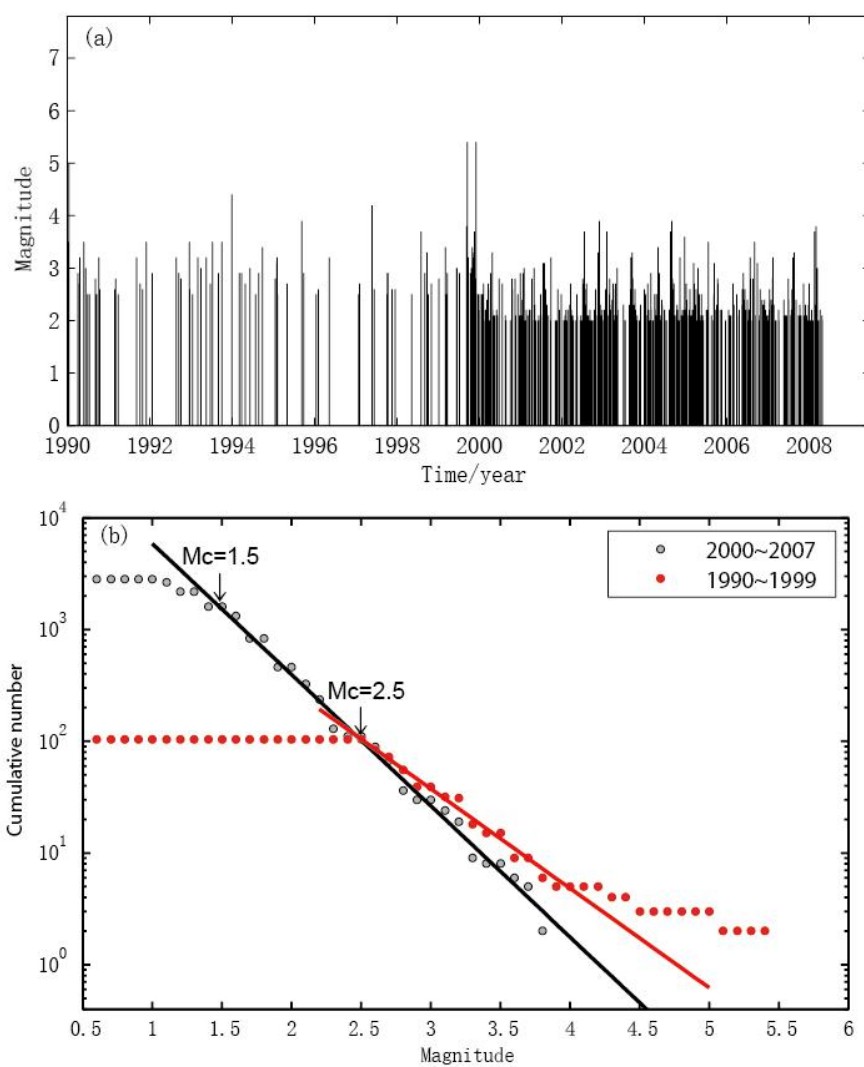

**Figure 2: (a)Magnitude as a function of time for earthquakes ($M_L \geq 2.0$) occurring in the study region. (b) Cumulative number vs. magnitude for earthquakes in the study region.**

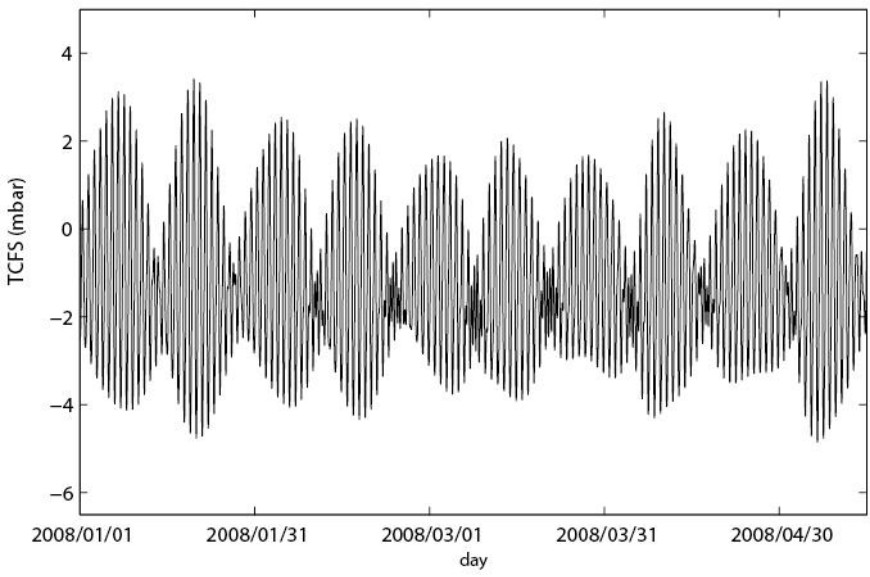

405

**Figure 3: Temporal variations of TCFS caused on the focal fault plane of the Wenchuan earthquake at a depth of 19 km.**

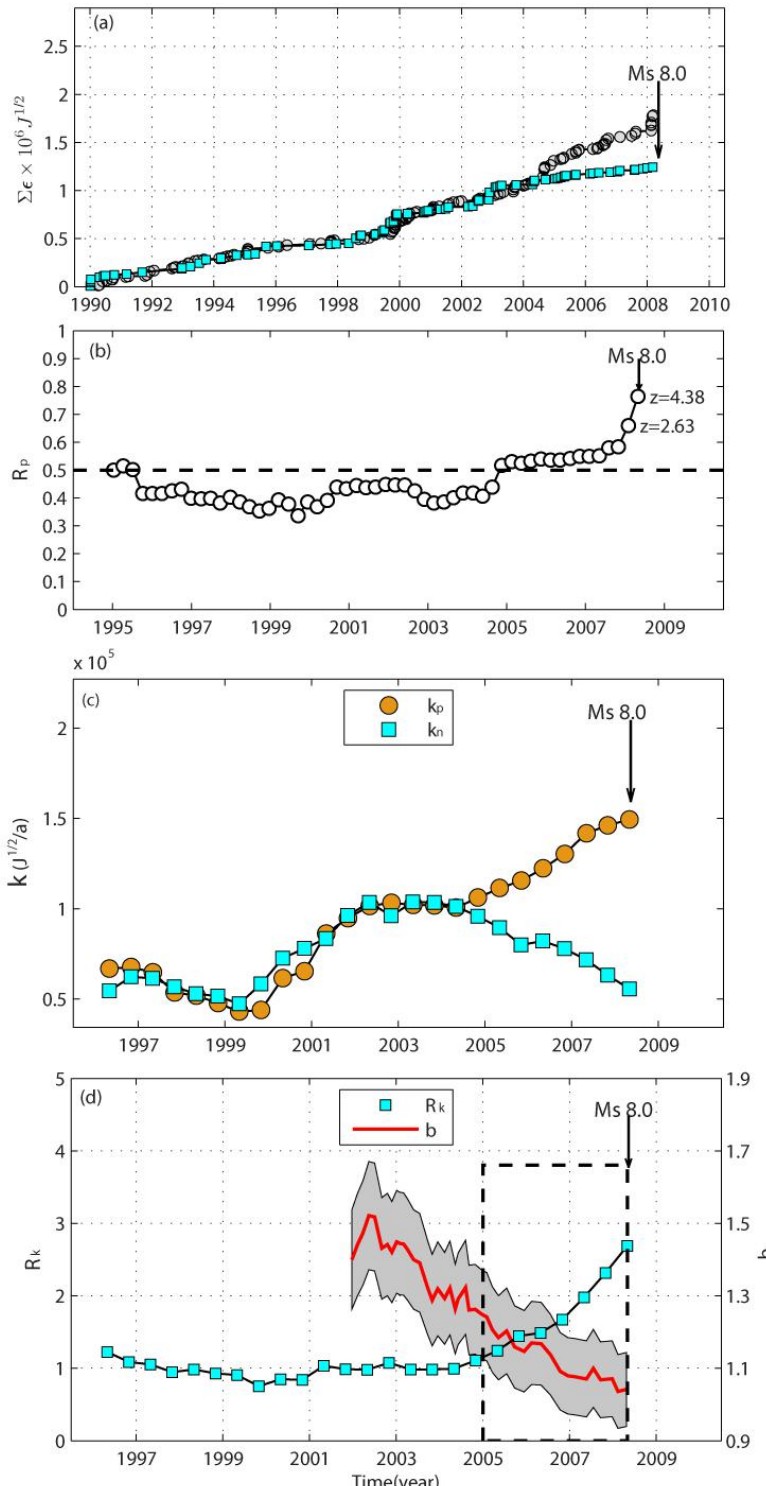

**Figure 4: (a) Cumulative seismic strain release curve. The line with "○" for PEQs, and the line with "□" for NEQs. (b) $R_p$ vs. time A moving 6-year time window moved by 6 months. (c) The time rate $k$ of CSSR vs. Time for both PEQs and NEQs. The orange circle shows the time rate $k$ for PEQs and the cyan square for NEQs. A moving 6-year time window moved by 6 months. (d) $R_k$ (cyan square) and b value(red line) as a function of time. The grey area indicates the 95% confidence limit of b value. The downward arrow shows the occurrence of the Wenchuan earthquake.**

**Supplementary material**

**PEQs (ΔTCFS>0) and NEQs (ΔTCFS<0) with seismic strain release $\varepsilon$ and ΔTCFS**

| No. | Date | Lat. (˚N) | Lon. (˚E) | $\varepsilon$ (×10$^4$) | ΔTCFS (Pa/m) | No. | Date | Lat. (˚N) | Lon. (˚E) | $\varepsilon$ (×10$^4$) | ΔTCFS (Pa/m) |
|---|---|---|---|---|---|---|---|---|---|---|---|
| 1 | 1990-1-4 | 30.87 | 103.40 | 0.756 | -0.16 | 110 | 2000-4-23 | 30.82 | 103.27 | 2.436 | 0.54 |
| 2 | 1990-1-9 | 30.72 | 103.22 | 3.600 | -0.22 | 111 | 2000-6-22 | 31.02 | 103.33 | 0.918 | 0.71 |
| 3 | 1990-1-9 | 30.67 | 103.18 | 1.357 | -0.22 | 112 | 2000-7-22 | 31.48 | 104.08 | 0.756 | 0.33 |
| 4 | 1990-1-9 | 30.70 | 103.22 | 2.004 | -0.23 | 113 | 2000-10-14 | 31.60 | 103.78 | 0.918 | -0.40 |
| 5 | 1990-4-6 | 30.58 | 103.32 | 1.116 | 0.30 | 114 | 2000-10-17 | 31.03 | 103.42 | 0.622 | -0.28 |
| 6 | 1990-4-20 | 30.58 | 103.28 | 0.756 | 0.16 | 115 | 2000-11-10 | 30.87 | 103.30 | 0.918 | -0.64 |
| 7 | 1990-4-23 | 30.63 | 103.28 | 2.004 | -0.50 | 116 | 2000-12-5 | 31.52 | 104.05 | 0.511 | -0.19 |
| 8 | 1990-5-28 | 30.57 | 103.23 | 3.600 | 0.28 | 117 | 2000-12-29 | 31.53 | 104.10 | 1.116 | 0.54 |
| 9 | 1990-6-13 | 31.48 | 104.07 | 1.357 | -1.42 | 118 | 2001-1-21 | 31.12 | 103.77 | 0.918 | 0.90 |
| 10 | 1990-6-30 | 31.18 | 103.62 | 0.511 | -0.24 | 119 | 2001-1-21 | 31.12 | 103.77 | 1.116 | 0.38 |
| 11 | 1990-7-16 | 30.92 | 103.12 | 0.511 | 0.12 | 120 | 2001-1-22 | 31.12 | 103.75 | 0.511 | 0.65 |
| 12 | 1990-9-7 | 31.53 | 103.98 | 0.918 | 0.31 | 121 | 2001-1-29 | 30.92 | 103.10 | 1.357 | 0.54 |
| 13 | 1990-9-19 | 31.32 | 103.82 | 0.511 | 0.27 | 122 | 2001-4-10 | 30.82 | 103.12 | 0.918 | 0.20 |
| 14 | 1990-10-7 | 30.70 | 103.22 | 2.004 | 0.42 | 123 | 2001-4-28 | 30.72 | 103.48 | 1.357 | -0.48 |
| 15 | 1990-10-16 | 31.48 | 103.98 | 0.622 | -0.48 | 124 | 2001-6-25 | 31.38 | 103.82 | 0.511 | -0.34 |
| 16 | 1991-2-27 | 31.62 | 104.07 | 0.622 | 0.97 | 125 | 2001-7-18 | 31.52 | 103.97 | 1.649 | 1.10 |
| 17 | 1991-3-7 | 30.88 | 103.37 | 0.918 | -0.93 | 126 | 2001-7-29 | 31.22 | 103.30 | 1.649 | -0.14 |
| 18 | 1991-3-29 | 30.68 | 103.53 | 0.511 | 0.63 | 127 | 2001-8-9 | 31.50 | 104.07 | 0.622 | 0.60 |
| 19 | 1991-9-9 | 31.00 | 103.37 | 2.004 | -0.32 | 128 | 2001-9-24 | 30.92 | 103.33 | 2.004 | 0.18 |
| 20 | 1991-10-9 | 31.43 | 103.97 | 0.756 | 0.83 | 129 | 2001-11-24 | 31.22 | 103.27 | 0.622 | 0.63 |
| 21 | 1991-10-31 | 31.42 | 103.97 | 0.622 | 0.63 | 130 | 2001-12-25 | 31.17 | 103.35 | 0.511 | 0.28 |
| 22 | 1991-12-1 | 31.03 | 103.42 | 3.600 | 0.94 | 131 | 2002-3-19 | 31.43 | 103.78 | 0.756 | -0.88 |
| 23 | 1992-1-24 | 30.82 | 103.43 | 1.116 | 0.34 | 132 | 2002-5-12 | 31.33 | 103.43 | 0.511 | -0.35 |
| 24 | 1992-8-24 | 31.32 | 103.62 | 2.004 | 0.73 | 133 | 2002-7-5 | 31.47 | 104.05 | 0.918 | 0.38 |

| 25 | 1992-9-16 | 31.18 | 104.18 | 1.116 | 0.03 | 134 | 2002-7-13 | 31.57 | 104.08 | 0.756 | 0.64 |
|---|---|---|---|---|---|---|---|---|---|---|---|
| 26 | 1992-10-5 | 31.12 | 104.02 | 0.918 | 0.28 | 135 | 2002-7-17 | 31.62 | 104.02 | 5.318 | -0.37 |
| 27 | 1992-12-18 | 31.02 | 103.63 | 3.600 | -0.75 | 136 | 2002-7-23 | 31.03 | 103.93 | 0.622 | -1.95 |
| 28 | 1992-12-20 | 31.13 | 103.07 | 0.622 | -0.43 | 137 | 2002-8-6 | 31.50 | 104.25 | 0.756 | 0.79 |
| 29 | 1992-12-22 | 31.33 | 103.88 | 0.511 | 0.72 | 138 | 2002-10-25 | 31.77 | 103.78 | 0.622 | 0.73 |
| 30 | 1993-1-16 | 31.20 | 103.92 | 0.511 | 0.98 | 139 | 2002-11-16 | 31.07 | 103.68 | 0.511 | -0.48 |
| 31 | 1993-3-2 | 30.57 | 103.42 | 2.004 | 0.14 | 140 | 2002-11-17 | 31.07 | 103.62 | 2.436 | 0.32 |
| 32 | 1993-3-2 | 30.68 | 103.15 | 0.756 | 0.22 | 141 | 2002-11-27 | 30.97 | 103.37 | 7.857 | -1.15 |
| 33 | 1993-3-30 | 31.25 | 103.87 | 1.357 | -0.37 | 142 | 2003-1-21 | 31.45 | 103.93 | 0.622 | 0.96 |
| 34 | 1993-5-17 | 30.57 | 103.22 | 2.004 | 0.08 | 143 | 2003-2-2 | 31.47 | 103.92 | 5.318 | -1.53 |
| 35 | 1993-6-22 | 31.18 | 103.57 | 0.756 | 0.70 | 144 | 2003-2-24 | 31.07 | 103.52 | 0.511 | -0.88 |
| 36 | 1993-7-11 | 31.48 | 103.93 | 3.600 | -0.13 | 145 | 2003-2-27 | 31.22 | 103.32 | 0.918 | 0.76 |
| 37 | 1993-8-31 | 31.38 | 104.02 | 1.116 | 0.31 | 146 | 2003-3-30 | 31.07 | 103.53 | 0.756 | 0.61 |
| 38 | 1993-9-30 | 31.67 | 103.73 | 3.600 | -0.05 | 147 | 2003-3-31 | 31.07 | 103.57 | 0.622 | -0.10 |
| 39 | 1994-3-5 | 31.50 | 103.93 | 1.116 | 0.81 | 148 | 2003-4-20 | 31.43 | 104.03 | 0.756 | -0.78 |
| 40 | 1994-3-24 | 31.07 | 103.88 | 0.756 | -0.27 | 149 | 2003-5-6 | 31.33 | 103.80 | 1.357 | 0.94 |
| 41 | 1994-3-24 | 31.07 | 103.88 | 1.116 | -0.37 | 150 | 2003-8-31 | 30.82 | 103.42 | 0.622 | 0.75 |
| 42 | 1994-4-28 | 31.27 | 103.43 | 0.756 | 0.18 | 151 | 2003-9-3 | 31.10 | 103.47 | 2.004 | 0.14 |
| 43 | 1994-6-7 | 31.17 | 103.83 | 1.357 | 0.58 | 152 | 2003-9-15 | 31.13 | 103.42 | 2.436 | 0.47 |
| 44 | 1994-7-26 | 31.32 | 103.75 | 0.511 | 0.16 | 153 | 2003-9-19 | 30.88 | 103.68 | 0.918 | 0.80 |
| 45 | 1994-8-23 | 31.58 | 104.07 | 1.116 | 0.18 | 154 | 2003-10-8 | 30.98 | 103.42 | 0.622 | -0.95 |
| 46 | 1994-9-26 | 31.58 | 104.12 | 2.961 | -0.50 | 155 | 2004-1-1 | 31.23 | 103.80 | 0.622 | 0.78 |
| 47 | 1995-1-17 | 31.47 | 103.62 | 0.918 | 0.46 | 156 | 2004-1-10 | 31.00 | 103.53 | 0.511 | 0.68 |
| 48 | 1995-2-1 | 30.95 | 103.47 | 1.649 | 0.90 | 157 | 2004-1-31 | 31.25 | 103.43 | 0.756 | 0.26 |
| 49 | 1995-2-3 | 31.00 | 103.47 | 1.116 | 0.25 | 158 | 2004-2-27 | 31.55 | 104.03 | 1.116 | 0.57 |
| 50 | 1995-2-3 | 31.00 | 103.47 | 2.004 | 0.07 | 159 | 2004-4-1 | 31.00 | 103.67 | 0.511 | 0.73 |
| 51 | 1995-2-3 | 31.00 | 103.47 | 0.511 | 0.42 | 160 | 2004-4-17 | 30.77 | 103.28 | 0.511 | -0.06 |
| 52 | 1995-2-7 | 31.07 | 103.90 | 0.511 | -0.61 | 161 | 2004-4-30 | 31.07 | 103.50 | 1.357 | 0.41 |

| | | | | | | | | | | | |
|---|---|---|---|---|---|---|---|---|---|---|---|
| 53 | 1995-5-3 | 31.58 | 103.93 | 0.756 | -0.08 | 162 | 2004-5-1 | 31.33 | 103.82 | 2.961 | -0.15 |
| 54 | 1995-9-9 | 31.38 | 103.87 | 7.857 | -0.67 | 163 | 2004-5-12 | 31.03 | 103.27 | 1.116 | -0.96 |
| 55 | 1995-9-27 | 31.57 | 104.12 | 1.116 | 0.96 | 164 | 2004-8-18 | 31.27 | 103.87 | 5.318 | 0.67 |
| 56 | 1996-1-14 | 31.47 | 103.93 | 0.511 | -0.80 | 165 | 2004-9-1 | 31.50 | 103.88 | 7.857 | 0.65 |
| 57 | 1996-2-3 | 30.82 | 103.13 | 0.622 | 0.04 | 166 | 2004-9-4 | 31.03 | 103.47 | 0.918 | 0.11 |
| 58 | 1996-5-9 | 31.23 | 103.53 | 2.004 | 0.58 | 167 | 2004-9-4 | 31.02 | 103.45 | 0.511 | 0.10 |
| 59 | 1997-1-24 | 30.85 | 103.50 | 0.511 | -1.33 | 168 | 2004-9-6 | 31.25 | 103.92 | 0.622 | 0.14 |
| 60 | 1997-2-1 | 30.70 | 103.37 | 0.756 | 0.10 | 169 | 2004-9-17 | 31.23 | 103.53 | 0.756 | -0.71 |
| 61 | 1997-6-13 | 31.17 | 104.22 | 0.622 | 0.65 | 170 | 2004-11-10 | 30.80 | 103.40 | 2.004 | 0.32 |
| 62 | 1997-10-5 | 30.57 | 103.28 | 0.511 | 0.16 | 171 | 2004-12-22 | 30.95 | 103.32 | 4.375 | 0.38 |
| 63 | 1997-10-7 | 31.45 | 103.92 | 1.116 | -1.06 | 172 | 2005-1-5 | 31.27 | 103.85 | 0.756 | -0.06 |
| 64 | 1997-10-11 | 31.27 | 103.72 | 1.116 | 0.14 | 173 | 2005-2-7 | 31.48 | 104.07 | 0.622 | -0.27 |
| 65 | 1997-10-13 | 31.27 | 103.22 | 1.116 | 0.05 | 174 | 2005-3-12 | 31.17 | 103.77 | 1.649 | 0.87 |
| 66 | 1997-11-18 | 31.25 | 103.33 | 0.622 | 0.60 | 175 | 2005-3-17 | 31.18 | 103.77 | 0.756 | -0.95 |
| 67 | 1997-12-14 | 31.58 | 104.12 | 0.622 | -0.59 | 176 | 2005-3-17 | 31.22 | 103.78 | 0.511 | -0.95 |
| 68 | 1998-5-11 | 31.28 | 103.73 | 0.511 | -1.31 | 177 | 2005-4-4 | 31.23 | 104.05 | 0.622 | 0.58 |
| 69 | 1998-8-2 | 30.68 | 103.37 | 5.318 | -0.87 | 178 | 2005-4-18 | 30.83 | 103.67 | 1.357 | 0.69 |
| 70 | 1998-9-3 | 31.50 | 104.00 | 1.116 | 0.35 | 179 | 2005-4-27 | 30.82 | 103.22 | 0.511 | -0.46 |
| 71 | 1998-9-22 | 31.57 | 104.07 | 2.436 | -0.36 | 180 | 2005-4-30 | 31.40 | 104.27 | 0.511 | -0.04 |
| 72 | 1998-10-5 | 30.75 | 103.32 | 0.511 | -0.91 | 181 | 2005-5-19 | 31.20 | 103.47 | 0.511 | -0.51 |
| 73 | 1998-11-2 | 31.25 | 103.73 | 0.756 | 0.08 | 182 | 2005-7-20 | 31.18 | 103.70 | 3.600 | 1.44 |
| 74 | 1999-1-5 | 31.50 | 103.98 | 0.918 | 0.84 | 183 | 2005-9-8 | 30.62 | 103.48 | 0.511 | -0.12 |
| 75 | 1999-3-6 | 31.18 | 103.35 | 2.961 | 0.43 | 184 | 2005-9-10 | 30.98 | 103.63 | 1.357 | 1.00 |
| 76 | 1999-3-13 | 31.33 | 104.30 | 0.511 | 0.58 | 185 | 2005-9-10 | 31.03 | 103.62 | 1.649 | 1.00 |
| 77 | 1999-3-23 | 31.28 | 104.33 | 1.116 | -1.00 | 186 | 2005-9-10 | 30.60 | 103.47 | 0.622 | -0.70 |
| 78 | 1999-6-13 | 31.32 | 103.27 | 1.357 | -0.37 | 187 | 2005-10-12 | 31.57 | 103.98 | 0.918 | 0.53 |
| 79 | 1999-6-16 | 31.25 | 103.82 | 1.357 | -1.97 | 188 | 2005-10-15 | 31.30 | 103.82 | 0.756 | 0.73 |
| 80 | 1999-6-16 | 31.23 | 103.37 | 0.622 | -1.94 | 189 | 2006-2-23 | 31.57 | 103.97 | 0.756 | -0.96 |

| 81 | 1999-7-7 | 30.80 | 103.25 | 1.116 | -0.01 | 190 | 2006-3-28 | 31.23 | 103.60 | 0.511 | 0.22 |
|---|---|---|---|---|---|---|---|---|---|---|---|
| 82 | 1999-9-8 | 30.63 | 103.47 | 6.464 | -0.95 | 191 | 2006-3-29 | 30.78 | 103.32 | 0.511 | -0.97 |
| 83 | 1999-9-14 | 31.60 | 104.07 | 0.756 | -0.74 | 192 | 2006-5-19 | 30.93 | 103.38 | 0.622 | 0.07 |
| 84 | 1999-9-14 | 31.58 | 104.07 | 0.511 | -0.76 | 193 | 2006-5-27 | 30.95 | 103.38 | 1.116 | 0.07 |
| 85 | 1999-9-15 | 31.62 | 104.07 | 0.511 | 0.54 | 194 | 2006-6-9 | 31.53 | 103.52 | 0.622 | -1.23 |
| 86 | 1999-9-23 | 31.32 | 103.78 | 2.004 | 0.23 | 195 | 2006-7-25 | 31.42 | 103.95 | 2.004 | 0.59 |
| 87 | 1999-10-12 | 30.85 | 103.47 | 1.116 | 0.49 | 196 | 2006-7-29 | 31.53 | 104.03 | 0.918 | 0.64 |
| 88 | 1999-10-20 | 31.60 | 104.08 | 1.357 | -0.20 | 197 | 2006-8-20 | 31.25 | 103.52 | 0.918 | 0.73 |
| 89 | 1999-10-28 | 31.58 | 104.07 | 2.961 | 0.15 | 198 | 2006-9-2 | 30.63 | 103.23 | 3.600 | 0.56 |
| 90 | 1999-11-7 | 31.48 | 103.53 | 0.622 | 0.60 | 199 | 2006-9-27 | 31.62 | 104.08 | 1.649 | 0.26 |
| 91 | 1999-11-9 | 30.95 | 103.43 | 0.511 | 0.95 | 200 | 2006-10-23 | 30.85 | 103.37 | 0.622 | -1.28 |
| 92 | 1999-11-10 | 30.73 | 103.55 | 2.004 | 0.98 | 201 | 2007-1-13 | 31.12 | 103.83 | 0.622 | -1.07 |
| 93 | 1999-11-10 | 30.73 | 103.55 | 0.511 | 0.88 | 202 | 2007-2-10 | 31.57 | 104.02 | 0.918 | -0.89 |
| 94 | 1999-11-12 | 30.73 | 103.55 | 2.436 | 0.92 | 203 | 2007-2-12 | 30.98 | 103.38 | 2.004 | 0.63 |
| 95 | 1999-11-16 | 30.58 | 103.52 | 5.318 | -0.14 | 204 | 2007-6-28 | 30.62 | 103.23 | 0.918 | 1.03 |
| 96 | 1999-11-21 | 31.57 | 104.03 | 0.918 | -0.81 | 205 | 2007-8-1 | 30.80 | 103.58 | 0.622 | -1.55 |
| 97 | 1999-12-1 | 31.18 | 103.83 | 0.622 | -0.01 | 206 | 2007-8-9 | 31.38 | 104.22 | 2.004 | 0.89 |
| 98 | 1999-12-3 | 30.75 | 103.57 | 0.511 | -0.67 | 207 | 2007-8-10 | 31.43 | 104.13 | 0.622 | -1.89 |
| 99 | 1999-12-20 | 31.22 | 103.32 | 0.511 | 0.82 | 208 | 2007-8-19 | 31.32 | 103.82 | 2.436 | 0.76 |
| 100 | 2000-1-5 | 31.63 | 104.05 | 0.511 | 1.08 | 209 | 2007-11-11 | 31.43 | 103.87 | 0.756 | -0.91 |
| 101 | 2000-1-22 | 31.60 | 104.03 | 0.511 | 1.08 | 210 | 2007-12-30 | 30.93 | 103.12 | 0.756 | -0.16 |
| 102 | 2000-2-2 | 31.62 | 104.07 | 0.511 | 0.41 | 211 | 2008-2-14 | 31.00 | 103.58 | 1.116 | 0.12 |
| 103 | 2000-2-5 | 31.33 | 103.92 | 0.756 | 0.06 | 212 | 2008-2-14 | 30.97 | 103.62 | 5.318 | 0.13 |
| 104 | 2000-3-18 | 30.83 | 103.52 | 0.622 | 0.17 | 213 | 2008-2-14 | 30.97 | 103.60 | 0.511 | 0.44 |
| 105 | 2000-3-18 | 30.85 | 103.53 | 0.756 | 0.19 | 214 | 2008-2-14 | 30.98 | 103.57 | 2.436 | 0.54 |
| 106 | 2000-3-18 | 30.83 | 103.53 | 0.511 | 0.19 | 215 | 2008-2-28 | 31.27 | 103.70 | 6.464 | 0.41 |
| 107 | 2000-4-5 | 31.27 | 103.58 | 0.622 | -0.14 | 216 | 2008-3-4 | 30.57 | 103.27 | 0.918 | -1.04 |
| 108 | 2000-4-9 | 30.93 | 103.60 | 1.116 | 0.64 | 217 | 2008-3-12 | 31.23 | 103.67 | 1.357 | 0.57 |

| 109 | 2000-4-22 | 31.63 | 104.03 | 0.918 | 0.97 | | | | | | | |

Note: Lat-Latitude; Lon-Longitude; $\varepsilon$- seismic strain release(measured in $J^{1/2}$ ). $\Delta$TCFS is measured in Pa per minute.

420