# Peer review of "Analysis of seismic strain release related to the tidal stress preceding the 2008 Wenchuan earthquake"

_Natural Hazards and Earth System Sciences, 2021_

## Referee Comment (RC1)

1. Whether Rp get stabilized after the occurrence of earthquake have to be included in the manuscript, since the authors have discussed only one major earthquake.
2. In page number 13, line number 246, it was mentioned that Fig. 5d, but it is not available in the manuscript. It should be Fig. 4d.
3. The author needs to explain the following, otherwise the manuscript merely observed the changes in the release of seismic release pattern.
    a. In page number 8, line number 164 – 167, the author has mentioned two categories of earthquakes, PEQs and NEQs. The author needs to provide table showing PEQs and NEQs with corresponding tidal stress and seismic strain release, either in manuscript or as a supplementary file.
    b. In page number 11 & 12, line number 220 – 225, the author has mentioned that seismic strain accelerated when kp and kn increases and it is mentioned that kn start decreasing from 2005 onwards, Why and how NEQs inhibits the release of strain has to explained.

---

## Community Comment (CC1)

1. Whether Rp get stabilized after the occurrence of earthquake.
2. In page number 13, line number 246, it was mentioned that Fig. 5d, but it is not available in the manuscript. It should be Fig. 4d.
3. The author needs to explain the following, otherwise the manuscript merely observed the changes in the release of seismic release pattern.
    a. In page number 8, line number 164 – 167, the author has mentioned two categories of earthquakes, PEQs and NEQs. The author needs to explain why positive and negative characteristics are shown by earthquakes, whether they have any spatial relation?
    b. In page number 9, line number 183 – 186, the author has only observed divergence between PEQs and NEQs from September 2004, need to explain the significance of divergence and transformation into seismic strain release.
    c. In page number 11 & 12, line number 220 – 225, the author has mentioned that seismic strain accelerated when kp and kn increases, but it is mentioned that kn start decreasing from 2005 onwards, Why the seismic strain release of NEQs decreases from 2005 and why NEQs went out of phase with PEQs are need to be explained.

---

## Community Comment (CC8)

1. Whether Rp get stabilized after the occurrence of earthquake have to be included in the manuscript, since the authors have discussed only one major earthquake.

Reply: We calculated the proportion $R_p$ of the seismic strain release for PEQs ($2.5 \leq M_L \leq 4.0$) that occurred from 2000 to 2021, applying a moving 5-year time window moved by 3 months. The result is shown below in the figure.

[Figure]

$R_p$ vs. time A moving 5-year time window moved by 3 months.

2. In page number 13, line number 246, it was mentioned that Fig. 5d, but it is not available in the manuscript. It should be Fig. 4d.

    Reply: We will revise it.

3. The author needs to explain the following, otherwise the manuscript merely observed the changes in the release of seismic release pattern.

a. In page number 8, line number 164 – 167, the author has mentioned two categories of earthquakes, PEQs and NEQs. The author needs to provide table showing PEQs and NEQs with corresponding tidal stress and seismic strain release, either in manuscript or as a supplementary file.

  Reply: We will add those data in manuscript.

b. In page number 11 & 12, line number 220 – 225, the author has mentioned that seismic strain accelerated when kp and kn increases and it is mentioned that kn start decreasing from 2005 onwards, Why and how NEQs inhibits the release of strain has to explained.

    Reply: When the stress in the focal region is close to a critical condition to release a large rupture, the tidal stress could take effect on the occurrence of earthquakes. The increasing tidal stress will promote the occurrence of earthquakes, making kp increase. While the decreasing tidal stress will inhibit the occurrence of earthquakes, making $k_n$ decrease.    So, it is the decreasing tidal stress to inhibit the release of strain for NEQs.

---

## Author Comment (AC1)

We must express our sincere thanks to Venkatanathan、Andrew Delorey and Anonymous referee#1 for their warmhearted help,their questions and suggestions.These questions and suggestions deserve consideration. Because the basic cause for earthquakes is still open to discussion, it's a little hard to reply some of these questions. In spite of this,we will do our best to give some explanations to them.

**Responce to Venkatanathan**

**Comments::**Herewith I have enclosed my comments on paper titled "Analysis of seismic strain releases related to tidal stress before the 2008 Wenchuan earthquake". Overall the authors have made observation that there is a seismic strain release, but the authors have to explain further why such behaviourial changes oberved between PEQs and NEQs.

**Reply:** Some researches reported the tidal triggering of earthquakes prior to moderate to large earthquakes (Tanaka et al., 2002b; Tanaka, 2010, 2012; Li et al., 2018). They investigated the tidal triggering of earthquakes in terms of event count. We investigated the tidal triggering of earthquakes before the Wenchuan earthquake in terms of seismic strain release, but not event count. Therefore the behaviourial changes oberved between PEQs and NEQs result from the tidal triggering of earthquakes prior to the Wenchuan earthquake indeed. The tidal triggering of earthquakes focus on the promoting effect of the tidal stresses, but our results reveal not only the promoting effect, but also the inhibiting effect of the tidal stresses. The increasing tidal stress will promote the occurrence of earthquakes, while the decreasing tidal stress will inhibit the occurrence of earthquakes when a large earthquake is impending.

**Comments:**Dear authors, I agree you have worked on seismic strain release. You have observed behaviourial changes of PEQs and NEQs , but there is no explanation that why NEQs decreases prior to the main event compared to the PEQs. You need to explain, when NEQs decreases due to decreasing tidal stress, then why PEQs

increases.

**Reply:** The Earth tide produces cyclic stress variations in the Earth. These stress variations, of the order of 1000~10000 Pa, are far smaller than the tectonic stress. When the stress in the focal region is at lower values, the tidal stress can not influence the occurrence of earthquakes, but when it is close to a critical condition to release a large rupture, the tidal stress could take effect on the occurrence of earthquakes. The tidal stress increase will promote the occurrence of earthquakes, making the seismic strain release accelerate for PEQs (corresponding to the increase of k in Fig.4c), and when the tidal stress decrease will inhibit the occurrence of earthquakes, making the seismic strain release decelerate for NEQs (corresponding to the decrease of k in Fig.4c).

**Comments:** Please include this explanation in the manuscript. For future studies - try to correlate PEQs and NEQs with dip (rake) of the fault.

**Reply:** Thank you for your comments and suggestions !We will do.

**Responce to Anonymous Referee #1**

**Comments:** Whether Rp get stabilized after the occurrence of earthquake have to be included in the manuscript, since the authors have discussed only one major earthquake.

**Reply:** We calculated the proportion $R_p$ of the seismic strain release for PEQs (2.5≤$M_L$≤4.0) that occurred from 2000 to 2021, applying a moving 6-year time window moved by 6 months. The result is shown below in the figure.

[Figure]

**Figure 1.1**

$R_p$ vs. time A moving 6-year time window moved by 6 months.

**Comments:** In page number 13, line number 246, it was mentioned that Fig. 5d, but it is not available in the manuscript. It should be Fig. 4d.

`Reply:` We will revise it.

**Comments:** The author needs to explain the following, otherwise the manuscript merely observed the changes in the release of seismic release pattern.

a. In page number 8, line number 164 – 167, the author has mentioned two categories of earthquakes, PEQs and NEQs. The author needs to provide table showing PEQs and NEQs with corresponding tidal stress and seismic strain release, either in manuscript or as a supplementary file.

`Reply:` We will add those data in manuscript.

b. In page number 11 & 12, line number 220 – 225, the author has mentioned that seismic strain accelerated when kp and kn increases and it is mentioned that kn start decreasing from 2005 onwards, Why and how NEQs inhibits the release of strain has to explained.

**Reply:** When the stress in the focal region is close to a critical condition to release a large rupture, the tidal stress could take effect on the occurrence of earthquakes.The increasing tidal stress will promote the occurrence of earthquakes, making kp increase.While the decreasing tidal stress will inhibit the occurrence of earthquakes, making $k_n$ decrease. So,it is the decreasing tidal stress to `inhibit the release of strain for NEQs.`

**Responce to Andrew Delorey**

**Comments:** The analysis shows results for PEQ (positive earthquakes) defined as when ΔTCFS >0. Did you also do the analysis on PEQ defined as TCFS>0? If so, what were the results? Can you discuss how the choice of PEQ impacts your interpretation and the underlying physics? I think the observation is pretty robust, but

there is a lot more analysis you could provide regarding your interpretation of the underlying physics and earthquake processes. As it stands, it is simply an interesting observation.

**Reply:** The results for PEQ defined as when TCFS >0 are shown in Figure1, quite different from that in our manuscript. It can be found in Figure1a that the CSSR curve for NEQs was over that for PEQs after 1991, and the difference was getting larger and larger. We also find a very slight change in the CSSR for PEQs but a larger one for NEQs. This result is difficult to understand. A simple stress model for a focal fault is showed in Figure 2, where $\tau_0$ (blue dotted line) denotes the tectonic stress acting on the focal fault, the black curve shows TCFS, the grey area shows the fluctuations of net stress on the focal fault. From point B to D or F to G $\Delta$TCFS >0, the net stress increases. From point A to B or D to F $\Delta$TCFS < 0, the net stress decreases. When the tectonic stress is close to a critical condition to release a large rupture, the increasing TCFS ($\Delta$TCFS >0) could promote the occurrence of earthquakes, but the decreasing TCFS ($\Delta$TCFS < 0) could take the opposite effect. Therefore the result obtained in our manuscript is easier to understand.

From point C to E TCFS >0, the net stress does not change totally. From point A to C or E to G TCFS < 0, the net stress does not change totally too. But why is the CSSR for NEQs larger that that for PEQs when PEQ is defined as when TCFS>0? Figure 3 shows the temporal variations of TCFS caused on the focal fault plane of the Wenchuan earthquake from 1 May 2008 to 12 May 2008. It can be found that TCFS stays below zero much longer than above it. Maybe this is the reason.

[Figure]

[Figure]

**Figure 1**

    **(a)** Cumulative seismic strain release curve. The line with "○" for PEQs, and the line with "□"

for NEQs. **(b)** $R_p$ vs. time. A moving 6-year time window moved by 6 months. **(c)** The time rate $k$

of CSSR vs. Time for both PEQs and NEQs. The orange circle shows the time rate $k$ for PEQs

and the cyan square for NEQs. A moving 6-year time window moved by 6 months. **(d)** $R_k$ (cyan

square) and b value(red line) as a function of time. The grey area indicates the 95% confidence limit of b value. The downward arrow shows the occurrence of the Wenchuan earthquake.

[Figure]

**Figure 2** A simple stress model for a focal fault

[Figure]

**Figure 3** Temporal variations of TCFS caused on the focal fault plane of the Wenchuan earthquake from 1 May 2008 to 12 May 2008.

**Comments:** Can you resolve any changes in behavior with shorter time resolution? You average over 5-years. Is there any change in behavior that you can resolve within the period of time when Rk is increasing?

**Reply:** Firstly ,we must correct a mistake. We averaged over 6 years, not 5 years in the manuscript. Now, we averaged over 3 years, four years and five years respectively, obtained the following results(Fig.4-6). Although these results are a little different from that in the manuscript, it is not enough to overturn conclusion obtained

previously. If we take a shorter time window, the data will reduce or become fewer, the results could be more uncertain. But it is worth noting that when we average over three years, the result shows that the **CSSR for PEQ accelerated from the beginning of 2008**(four months and more before the Wenchuan event).

[Figure]

**Figure 4**

**(c)** The time rate $k$ of CSSR vs. Time for both PEQs and NEQs. The orange circle shows the time rate $k$ for PEQs and the cyan square for NEQs. A moving 5-year time window moved by 6 months. **(d)** $R_k$ (cyan square) and b value(red line) as a function of time. The grey area indicates the 95% confidence limit of b value. The downward arrow shows the occurrence of the Wenchuan earthquake.

[Figure]

**Figure 5**

**(c)** The time rate $k$ of CSSR vs. Time for both PEQs and NEQs. The orange circle shows the time rate $k$ for PEQs and the cyan square for NEQs. A moving 4-year time window moved by 6 months.

**(d)** $R_k$ (cyan square) and b value(red line) as a function of time. The grey area indicates the 95% confidence limit of b value. The downward arrow shows the occurrence of the Wenchuan earthquake.

[Figure]

**Figure 6**

**(c)** The time rate $k$ of CSSR vs. Time for both PEQs and NEQs. The orange circle shows the time rate $k$ for PEQs and the cyan square for NEQs. A moving 3-year time window moved by 6 months.

**(d)** $R_k$ (cyan square) and b value(red line) as a function of time. The grey area indicates the 95% confidence limit of b value. The downward arrow shows the occurrence of the Wenchuan earthquake.

**Comments:** Why should $k_n$ decrease when approaching the Wenchuan earthquake? It seems more intuitive that both $k_p$ and $k_n$ should increase, even if $k_p$ increases faster.

**Reply:** Figure 7a shows the epicentral distribution of PEQs and NEQs ($M_L \geq 3.0$) that occurred along the Longmenshan fault from Jan, 2005 to Apr. 2008. Only PEQs (red circles)) occurred in the study region.    Figure 7b and 7c show magnitude as a function of time for NEQs and PEQs ($M_L \geq 3.0$) that occurred in the study region, respectively. It is found that from Jan. 2005 to Apr. 2008 there was no $M_L \geq 3.0$ NEQ, but more $M_L \geq 3.0$ PEQs than before. That might be the reason that $k_n$ decreased when the Wenchuan earthquake was approaching, i.e., the tidal stress decrease inhibited the occurrence of earthquakes. It is found not only from Figure 4c in the manuscript but also Figure 4-6 in this reply that $k_n$( cyan squares) shows a decreasing trend from Jan. 2005 to the time of the occurrence of the Wenchuan earthquake. When a shorter time window is applied the change of $k_p$ become a little more complicated, but the difference between $k_p$ and $k_n$ is still significant after Jan. 2005.

[Figure]

**Figure 7**

(a)   Epicentral distribution of PEQs (red circles) and NEQs (blue squares) ($M_L \geq 3.0$) that occurred along the Longmenshan fault from Jan, 2005 to Apr. 2008.   (b) Magnitude as a function of time for NEQs ($M_L \geq 3.0$). (c) Magnitude as a function of time for PEQs ($M_L \geq 3.0$)

**Comments:** There is another change in behavior around 1999, which can be seen in Figure 4a and 4c. Is there any explanation for that? Is it related to the change in instrumentation?

**Reply:** Two M5.4 earthquakes occurred on 14 September and 30 November 1999 respectively. The change in behavior around 1999 was caused by their aftershocks.

**Comments:** Figure 1b, what component of strain are you showing?

**Reply:** When a focal fault ruptures to cause an earthquake, strain energy will be released. A portion of it is released through seismic waves, i.e. seismic energy Es. Strain in this study is obtained by taking the square root of $E_S$, called as the Benioff strain in seismology. The strain energy is related to the shear deformation of the focal fault, hence strain here has the significance of shear strain.

**Comments:** You write, "as the length of time with ΔTCFS>0 is almost the same as that with ΔTCFS<0". What do you mean by almost? Do you account for the difference in your analysis? You should compare observed versus actual expected, not observed versus "almost" expected. This could impact your results.

**Reply:** TCFS is the resultant stress produced by earth tides of different periods。 Its change over time is not always regular. So, the length of time with ΔTCFS>0 is not usually the same as that with ΔTCFS<0", but the difference is slight (see Figure 2 and Figure 3). The word "almost" is not suitable to use here. Wei will change it to "as the length of time with ΔTCFS>0 is approximately equal to that with ΔTCFS<0" .

**Comments:** There are some minor language problems, that could be fixed by having the manuscript reviewed by a more experienced English writer.

**Reply:** We will have the revised manuscript polished.

---

## Author Response (AR2)

**Responce**

We sincerely thank the reviewers for their recommendations and comments.

Comments:

I have made a few recommended revisions in the attached document.
Reply :revised.

I recommend adding Figure 8 from the response to reviewer document, which shows the geographic and temporal distribution of PEQ and NEQ earthquakes, to the manuscript. Particularly the geographic distribution is worth showing to the readers.
Reply :added.

The results from TCFS and (delta)TCFS suggest that earthquakes preferentially occur when (delta)TCFS is positive and TCFS is negative. So, when stress rate is positive, but stress magnitude is still negative. This is a potentially interesting result worth pondering further. I don't think you need to do a full analysis of this point, but it is worth mentioning in this manuscript.
Reply :   When $\Delta$TCFS is positive, TCFS is negative or positive. We only suggested that earthquakes preferentially occur When $\Delta$TCFS is positive, but this phenomenon occurs when the focal medium is close to a critical state. If we let $\tau_0$ denote the tectonic stress, without considering other factors the total stress at a focal fault $\tau=\tau_0+$TCFS. If $\tau_0$ does not change, when TCFS increase $\tau$ increases, and vice versa. When $\Delta$TCFS is positive, though TCFS is negative, but $\tau$ increases from the lowest.

The authors state this but perhaps should emphasize it more, that kp and kn both fluctuate due to seismicity rate and perhaps for other reasons, but it is the separation that may be most important. That is, kp or kn, are not very informative by themselves.
Reply :We made a suitable revisions for this.